genetics, ecology, evolution

captive breeding, stocking, Atlantic salmon, lifetime reproductive success

**Author for correspondence:**
Ronan James O'Sullivan
e-mail: ronan.j.osullivan@umail.ucc.ie

†These authors share first authorship.
‡Work conducted when based at the Division of Genetics and Physiology, Department of Biology, University of Turku, Turku, Finland.

# Captive-bred Atlantic salmon released into the wild have fewer offspring than wild-bred fish and decrease population productivity

Ronan James O'Sullivan[1,2,†], Tutku Aykanat[3,†,‡], Susan E. Johnston[4,‡], Ger Rogan[5], Russell Poole[5], Paulo A. Prodöhl[6], Elvira de Eyto[5], Craig R. Primmer[3,7,‡], Philip McGinnity[1,2,5] and Thomas Eric Reed[1,2]

[1]School of Biological, Earth and Environmental Sciences, University College Cork, Distillery Fields, North Mall, Cork, Ireland
[2]Environmental Research Institute, University College Cork, Cork, Ireland
[3]Organismal and Evolutionary Biology Research Program, Faculty of Biological and Environmental Sciences, University of Helsinki, PO Box 56, 00014 Helsinki, Finland
[4]School of Biological Sciences, University of Edinburgh, UK
[5]Marine Institute, Furnace, Newport, Mayo, Ireland
[6]Institute for Global Food Security, School of Biological Sciences, Medical Biology Centre, Queen's University Belfast, Belfast, UK
[7]Institute of Biotechnology, University of Helsinki, Helsinki, Finland

RJO, 0000-0003-3650-2048; TA, 0000-0002-4825-0231; SEJ, 0000-0002-5623-8902; PAP, 0000-0001-8570-9964; EDE, 0000-0003-2281-2491; CRP, 0000-0002-3687-8435; PM, 0000-0001-5199-5632; TER, 0000-0002-2993-0477

The release of captive-bred animals into the wild is commonly practised to restore or supplement wild populations but comes with a suite of ecological and genetic consequences. Vast numbers of hatchery-reared fish are released annually, ostensibly to restore/enhance wild populations or provide greater angling returns. While previous studies have shown that captive-bred fish perform poorly in the wild relative to wild-bred conspecifics, few have measured individual lifetime reproductive success (LRS) and how this affects population productivity. Here, we analyse data on Atlantic salmon from an intensely studied catchment into which varying numbers of captive-bred fish have escaped/been released and potentially bred over several decades. Using a molecular pedigree, we demonstrate that, on average, the LRS of captive-bred individuals was only 36% that of wild-bred individuals. A significant LRS difference remained after excluding individuals that left no surviving offspring, some of which might have simply failed to spawn, consistent with transgenerational effects on offspring survival. The annual productivity of the mixed population (wild-bred plus captive-bred) was lower in years where captive-bred fish comprised a greater fraction of potential spawners. These results bolster previous empirical and theoretical findings that intentional stocking, or non-intentional escapees, threaten, rather than enhance, recipient natural populations.

## 1. Introduction

The active management of populations to mitigate against anthropogenic change or increase opportunities for commercial or recreational exploitation occurs for many species [1–3]. Wild population management often incorporates captive breeding programmes, where reintroduction after extirpation [4,5] or supplementation of existing populations [6–8] are the conservation goals. However, evidence suggests that the deliberate (stocking) or accidental escape of captive-bred conspecifics may depress the productivity of wild populations through ecological [9,10], genetic [11–18] or epigenetic

mechanisms [19–21], as well as impacting other species [22], thus raising questions regarding the viability of wild populations that experience inputs of captive-bred individuals. Indeed, the lifetime fitness of released individuals relative to wild individuals is rarely directly measured.

The release of captively bred salmonine fish has been practised for over 150 years for the purposes of reintroduction, arresting population declines or providing increased opportunities for commercial harvest or recreational angling [23]. 'Sea ranching' refers to situations where captive-bred fish are released as smolts (the life stage at which fish are physiologically ready to enter the marine environment) and then captured on their return migration as adults, either in a fishery or for use as broodstock for the next captive-bred generation. Sea ranching programmes aim to recover all the fish released into the wild as adults as part of either a commercial fishery or an experimental scientific programme. This is in contrast with 'stocking', where hatchery-produced fish may be released as eggs, juveniles or smolts, and intentionally allowed to spawn naturally in the wild once they have returned to the rivers as adults. However, some ranched fish may be released intentionally or may escape inadvertently, thus affording them the opportunity to spawn in the wild.

Salmonine fishes experience a severe survival bottleneck in the wild, with average egg-to-smolt survival rates in Burrishoole Atlantic salmon ranging from 0.3 to 1.1% [24]. By contrast, the hatchery environment with its absence of predators, food ad libitum and disease prevention can lead to very high cumulative survival of captive-bred fish. Therefore, stocking and ranching can provide more fish for angling, and/or increase commercial catches [25]. However, the intentional stocking or inadvertent release of captive-bred fish into the wild may threaten the long-term viability of recipient wild populations [10,26], thus creating a vicious circle, whereby artificial propagation increases the population's reliance on future interventions. In the past, captive-bred fish were assumed to be ecologically equivalent to wild-bred fish. However, mounting evidence (electronic supplementary material, table S1) demonstrates that hatchery fish have lower survival post-release, relative to wild-bred fish [27–30] and are less likely to obtain and defend breeding sites or mates [13,16,31–35]. The longer a given individual spends in captivity, the more its phenotype diverges from that of wild-bred fish, and hence the worse its performance in the wild is expected to be, with a trade-off between higher cumulative survival and reduced wild performance post-release. Therefore, the numerical gains of stocking may be marginal or even negative, which argues against stocking even purely on demographic grounds [10,23,36–39]. The inferior post-release survival and spawning behaviour of captively bred fish, coupled with negative demographic consequences, raises two questions—does the poor performance of individual captive-bred fish translate into reduced overall fitness for the captive-bred group relative to wild-bred fish? If so, do spawning captive-bred fish affect population productivity?

Any potential short-term demographic benefits of stocking or ranching must be weighed against longer term impacts owing to transgenerational carry-over effects [20,26–28]. Experiments with Atlantic salmon *Salmo salar* L. have shown that the parental hatchery environment can affect the survivorship of wild-bred offspring [20], which can occur via genetic responses to domestication selection in captivity, or via non-genetic effects of the captive environment, including maternal effects and epigenetic inheritance. Classic studies [13,14,16] on steelhead trout, *Oncorhynchus mykiss* Walbaum, have demonstrated that even a few generations of captive-rearing can reduce the performance of captive-bred individuals and their offspring in the wild via genetic changes that occurred in captivity, even when broodstock were obtained from the local wild population [40,41]. Therefore, interbreeding between captive- and wild-bred fish entails evolutionary risks for the wild population, as introgression of 'hatchery alleles' can negatively affect the fitness of hybrids in the wild, potentially depressing population size or productivity [17].

Natural selection in the wild should select against wild-bred individuals with high levels of captive ancestry [42], which in turn should purge hatchery alleles. However, this purging process would still incur a demographic cost to the wild population [43], while continued influx of hatchery fish would lead to further introgression and fitness depression. We use lifetime reproductive success (LRS) data from the Burrishoole catchment in Ireland to (i) compare lifetime fitness in nature of wild- and captive-bred Atlantic salmon that had the opportunity to spawn naturally and (ii) quantify the resulting impacts on population productivity of annual intrusions of captive-bred fish. We then compare the LRS of captive-bred fish against that of wild-bred fish to test the hypothesis that hatchery-induced genetic or environmental effects reduced the fitness of captive-bred fish relative to wild-bred fish when both spawned naturally in the wild. Finally, we use a density-corrected measure of overall lifetime productivity (adult recruits per adult spawner) to test the prediction that population productivity is lower in years where captive-bred fish comprise a greater fraction of the potential spawning population.

## 2. Methods

### (a) Lifetime reproductive success data

An experimental ranching programme has operated in Burrishoole since the 1960s, where captive-bred fish are tagged and released as smolts. The ranching programme was established using primarily local broodstock, but also with the inclusion of non-local brood in the earlier years of the programme. In the Burrishoole system (see [36]), the majority (approx. 90%) of wild-bred fish migrate to sea at age 2+, a little over 2 years after hatching with a small fraction (approx. 10%) migrating as either age 1+ or age 3+ juveniles. Similarly, captive-bred individuals migrate at ages 1+ or 2+. Prior to release, captive-bred fish are microtagged and their adipose fin is clipped, so as to distinguish them from wild-bred fish upon their return as adults. Microtagging refers to the procedure of injecting a coded wire tag (a length of magnetized stainless steel wire 0.25 mm in diameter) into the nose of fish. The majority of captive- and wild-bred fish return from the ocean to breed after one full 'sea-winter' as so-called grilse, with the remainder returning as multi-sea-winter (MSW) fish [44]. Upon their return, captive-bred fish are either caught in a rod fishery, retained as broodstock, released up the catchment due to hatchery constraints or for enhancement/angling, under the assumption that these fish are recaptured before spawning. Most fish of both provenances return from the ocean during May to September, but do not spawn until late winter (November–January), with peak spawning in mid-December. Fish that survive spawning return to the sea early

in the following calendar year. Individual fish in our data that did not display the conventional 2+-grilse life cycle had their LRS appropriately indexed to the correct years across which they spawned (see lines 280–693 in Code text file available at http://data.marine.ie/geonetwork/srv/eng/catalog.search#/metadata/ie.marine.data:dataset.4346).

A total trapping system on the catchment has allowed for censusing and tissue sampling of all anadromous Burrishoole salmon that were potential spawners. Trapped fish are measured for fork-length, with scale samples taken for molecular parentage assignment and sexing. Whether a fish is caught pre-spawning (i.e. in the upstream traps on their return from the ocean) or post-spawning (i.e. in the downstream traps on their migration back to the ocean) was also recorded. 94.21% of wild-bred fish were tissue sampled as kelts in the downstream traps, while 94.74% of captive-bred fish were sampled pre-spawning in the upstream traps. Captive- and wild-bred salmon in Burrishoole have similar propensities for male precociality [45] so we do not believe their absence from our data biases our estimates of relative reproductive success (RRS), as our estimates of absolute LRS for each provenance are expected to be equally biased by failing to account for mature male parr. This trapping regime has allowed for the collection of annual census data based on total counts, phenotypic and genetic sampling of the potential anadromous wild-spawning population (electronic supplementary material, table S2) which, in combination with molecular parentage assignment, has facilitated the estimation of lifetime fitness for individual fish using a molecular pedigree. See [46,47] for details of tissue sampling procedures, identity and parentage analysis, sex determination and pedigree reconstruction. Pedigree data available at [48].

The discrepancy in when each provenance was sampled (upstream versus downstream) could bias LRS estimates downward for wild-spawning, captive-bred fish, due to the higher cumulative in-river mortality risk inflating their expected number of zero LRS records relative to wild-bred parents. We explored this by comparing non-zero LRS records between the provenances. The mean LRS was still significantly lower for captive-bred relative to wild-bred fish, demonstrating that this source of bias was not driving the main results (in qualitative terms) from the analyses. See electronic supplementary material, text S1 for further details.

## (b) Productivity data

We estimated annual productivity of the population as the average contribution of adult offspring to the next generation by fish that had an opportunity to breed in the wild (i.e. the combined number of wild- and captive-bred potential spawners). To quantify the total number of recruiting offspring produced by a spawning cohort we assumed, for simplicity, that all wild-bred fish smolted at age 2+ and then returned as adults after either one or two winters at sea. Therefore, the total recruits produced by spawning cohort $t$ was estimated by summing the number of wild-bred grilse returning in year $t+4$ and the number of wild-bred MSW fish returning in year $t+5$. This was then divided by the number of spawners (wild-bred + captive-bred) in year $t$, providing us with a ratio of adult recruits per spawner, giving us an adult-to-adult productivity measure. We estimated productivity for all years where complete life cycle data were available (43 years; 1970–2012). Since female MSW salmon and female grilse differ in their fecundity [49], variation in the annual grilse : MSW ratio could affect productivity if estimated using ova deposition. To explore this, we calculated an alternative productivity measure that involved 'converting' adults into eggs, using an unpublished relationship between female length and fecundity (Marine Institute 1992–2012, unpublished data; electronic supplementary material, text S2).

## (c) Relative reproductive success between provenances

Fitness, measured as LRS, was estimated by counting the number of adult fish returning in subsequent years that could be assigned genetically as offspring of a focal parent. Thus, LRS was an individual-level analogue of our population-level productivity (recruits per spawner) measure. RRS of each provenance was then estimated for each spawning cohort by dividing their arithmetic mean LRS by that of wild-bred fish. Thus, wild-bred fish have an RRS value of 1 in any given spawning cohort, and RRS values of less than 1 for captive-bred fish indicate lower fitness compared to wild-bred fish. Parentage assignment errors and incomplete sampling of parents and offspring were accounted for as per [50], yielding an unbiased estimator of RRS for each cohort (electronic supplementary material, table S3). The method of [50] involved: (i) subtracting the number of offspring successfully assigned back to a parent from the number of offspring sampled; (ii) dividing this difference by the total number of potential parents; (iii) multiplying by the pedigree-derived assignment error, divided by one minus the assignment error; and (iv) subtracting the result from the arithmetic mean LRS. This was done separately for each provenance, with the result being an unbiased estimate of the mean LRS for both wild- and captive-bred salmon. Dividing captive-bred mean LRS by wild-bred mean LRS yielded the unbiased RRS estimator. This allowed us to examine variation in the RRS of captive-bred fish across six cohorts (1977, 1978, 1980, 1981, 1985 and 1989). Separate one-tailed permutation tests were used to test for significant differences in the mean LRS between captive- and wild-bred salmon for each of the six cohorts. The permutation tests generated 1 000 000 estimates of the difference in arithmetic mean LRS between captive- and wild-bred fish. $p$-values for each test were calculated as the proportion of the permuted samples greater than the observed difference in the mean LRS between the provenances. One-tailed tests were chosen as we had an *a priori* expectation for lower LRS in captive-bred fish based on previous work [13,14,16]. Permuting the difference between the mean LRS estimates is mathematically equivalent to testing if the RRS of captive-bred fish is different from one—one being the relative fitness of wild-bred fish. To assess evidence for the hypothesis that, across all cohorts, there was an overall pattern of captive-bred salmon displaying lower LRS than wild-bred salmon, we combined the $p$-values from each of the six permutation tests using Fisher's combined probability test (FCPT). FCPT relies on taking the natural logarithms of the permuted $p$-values. In cases where the permuted $p$-value was zero, we used the 'independence_test' function in the 'coin' R package to derive a non-zero $p$-value for use in the FCPT [51,52]. When the permuted and 'coin'-derived $p$-values were on opposite sides of the arbitrary 0.05 significance threshold, the more conservative $p$-value was chosen. This analytical pipeline was also used to assess evidence for the hypothesis that captive-bred fish displayed lower LRS than wild-bred fish when data were separated by sex. Again, the sex-specific comparisons were done for each cohort separately, as well across all cohorts. For overall provenance and sex comparisons, the weighted geometric mean of relevant cohort-specific RRS estimates was calculated.

Lower LRS in captive-bred fish relative to wild-bred could be explained by reduced survival of their offspring in the wild due to transgenerational effects of the hatchery on offspring phenotypes [20] via genetic [41] or epigenetic inheritance [19]. However, captive-bred fish could simply have lower spawning success [32] or, in the case of females, higher rates of egg retention [49]. If that were the case, one would expect that a higher fraction of captive-bred fish would have an LRS equal to zero due to never having spawned. We thus tested whether captive-bred fish in our pedigree had a higher proportion of zero LRS than wild-bred fish using the 'prop.test' function in R. Having

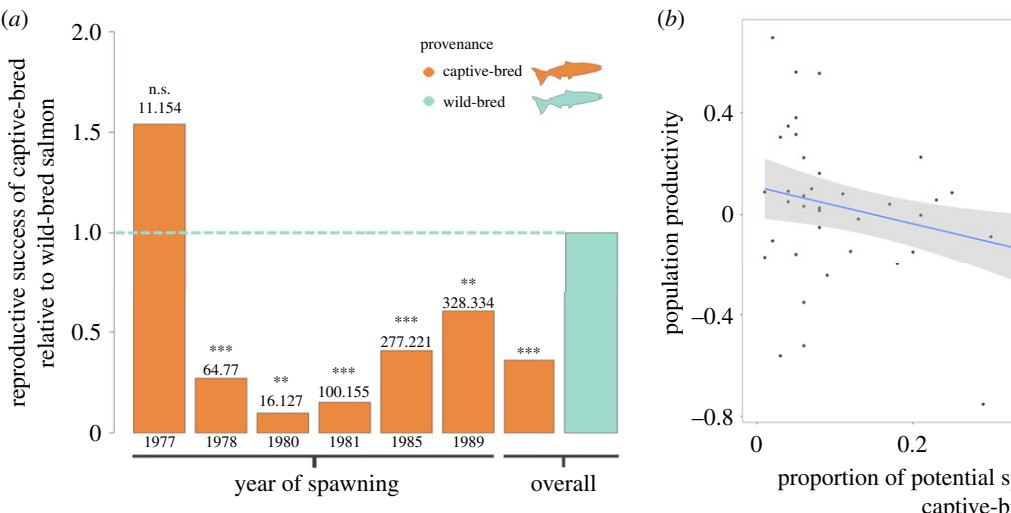

**Figure 1.** (a) Overall and cohort-specific comparisons of RRS for captive- and wild-bred Atlantic salmon in the Burrishoole catchment, Ireland. Overall RRS comparison estimated as the weighted geometric mean of the six cohort point estimates. Significance of the overall comparison determined using FCPT, where $X^2 =$ 117.94 with 12 degrees of freedom. Significance of cohort-specific comparisons was determined using one-tailed permutation tests. Horizontal line for emphasis of increase/decrease in reproductive success of captive-bred fish relative to wild-bred fish. Numbers on top of bars represent the number of captive-bred (left number) salmon and wild-bred (right number) salmon used in cohort-specific comparisons. $*p < 0.05$, $**p < 0.01$, $***p < 0.001$. (b) Productivity of the mixed population as a function of the annual proportion of potentially spawning fish that were captive-bred. The solid line represents the line-of-best fit from a linear model, and shading represents the 95% confidence interval. (Online version in colour.)

found a significantly higher proportion of zero LRS in captive-bred fish (electronic supplementary material, text S3), we then restricted our dataset to records where LRS > 0 and performed a one-tailed permutation test as above. A significant difference between captive-bred and wild-bred fish would be consistent with transgenerational effects of the hatchery environment on the survival of offspring of captive-bred parents. The fecundity of captive-bred females is approximately 1.4 times that of wild-bred females in the Burrishoole system, as captive-bred fish have a higher number of smaller eggs per kilogram of maternal bodyweight in comparison with wild-bred fish (Marine Institute 1970–2012, unpublished data). Thus, in the case of females, lower LRS, having excluded the zeros, for captive-bred fish relative to wild would be despite the fact that they can deposit more eggs per capita. Finally, we estimated the reduction in the mean individual LRS across the six cohorts relative to a hypothetical pure wild-bred population, as the result of intrusions by captive-bred fish. This was estimated by multiplying the number of captive-bred fish in a given cohort by their estimated RRS that year, doing the same for wild-bred fish, and then summing these products across cohorts and dividing by the grand total of captive-bred and wild-bred fish. This result was then subtracted from one and multiplied by 100 to give the percentage reduction in the mean individual LRS in the mixed population in the parental generation relative to a hypothetical pure population.

### (d) Effect of intrusions of captive-bred fish on population productivity

As a measure of hatchery intrusion, we used the proportion of the total number of returning wild- and captive-bred adults that had an opportunity to spawn in the wild, that were captive-bred. This figure ranged from 0.01 to 0.59. Population productivity was calculated as recruits per spawner, as explained above, with spawners indexed to year $t$, as per the hatchery influence measure. Regressing population productivity on proportion captive-bred fish would be problematic as it ignores potential density dependence, which can be strong in Atlantic salmon [53,54]. We approximated annual population density by the total annual return of fish, assuming the area of spawning and

rearing habitat within the catchment was relatively fixed across years. While this density measure was only poorly correlated with the annual proportion of captive-bred fish (Pearson's correlation: $r = 0.056$, $t = 0.36$, d.f. = 41, $p = 0.72$), failing to account for density dependence could still obscure the relationship between hatchery intrusion and population productivity. In fisheries science, nonlinear productivity relationships are typically assumed (e.g. Beverton–Holt or Ricker functions [55]). Rather than choosing an arbitrary stock–recruit function, we instead fit a generalized additive model (GAM) assuming Gaussian errors using the '*mgcv*' [56] R package, where the natural logarithm of population productivity was regressed onto total annual numbers of fish that had the opportunity to spawn in the wild, with the latter set as a smoothing term with nine knots (electronic supplementary material, figure S1). The residuals from this model are equivalent to a density-corrected productivity measure, as they represent a density-independent stock–recruitment relationship. Using a linear model, we then regressed these residuals against the proportion of captive-bred fish in a spawning cohort, with the prediction that population productivity would be lower in years of stronger hatchery intrusion. This analysis was repeated using our alternative 'ova-per-ovum' productivity measure and proportion ova contributed by captive-bred fish as the explanatory variable to check that the results were robust to converting adults into eggs (electronic supplementary material, text S2). Visual inspection of diagnostic plots showed that all model assumptions were met (electronic supplementary material, figures S2 and S3). All analyses were conducted in R version 3.6.1.

## 3. Results

One-tailed permutation tests revealed significantly reduced LRS for captive-bred compared to wild-bred fish in all but one of the six spawning cohorts (figure 1a and table 1). FCPT revealed an overall effect of reduced RRS in captive-bred fish across the six cohorts (FCPT: $X^2 = 117.94$, d.f. = 12, $p < 0.001$; figure 1a and table 1). For female-specific comparisons, captive-bred females displayed lower LRS than wild-

**Table 1.** Cohort- and sex-specific, and overall estimates of unbiased RRS for Atlantic salmon. For cohort- and sex-specific estimates, $p$-values were determined by one-tailed permutation tests. For overall comparisons, RRS was estimated using the weighted geometric mean of cohort- or sex-specific estimates of RRS, and a $p$-value determined using FCPT, assuming a $X^2$-distibution with 12 degrees of freedom. Significant $p$-values in bold.

| cohort | overall | | female | | male | |
|---|---|---|---|---|---|---|
| | RRS | *p*-value | RRS | *p*-value | RRS | *p*-value |
| 1977 | 1.54 | 0.76 | 0.21 | 0.14 | 20.92 | 0.99 |
| 1978 | 0.27 | **<0.001** | 0.15 | **<0.001** | 1.68 | 0.48 |
| 1980 | 0.10 | **<0.01** | 0.20 | 0.08 | 0.00 | 0.07 |
| 1981 | 0.15 | **<0.001** | 0.13 | **<0.001** | 0.17 | **<0.001** |
| 1985 | 0.41 | **<0.001** | 0.38 | **<0.001** | 0.42 | **<0.01** |
| 1989 | 0.61 | **<0.01** | 0.17 | **<0.001** | 1.28 | 0.72 |
| overall | 0.36 | **<0.001** | 0.30 | **<0.001** | 0.67 | **<0.001** |

bred females in all six cohorts, with the reduction being significant in four cohorts (electronic supplementary material, figure S4). RRS of captive-bred females ranged from 0.13 to 0.38, with an average RRS of 0.30 ($X^2 = 107.67$, d.f. = 12, $p < 0.001$; table 1). Captive-bred males displayed lower LRS than wild-bred males in three of the six cohorts, with the reduction being significant in two cohorts (electronic supplementary material, figure S4). RRS of captive-bred males ranged from 0 to 20.92, with an average RRS of 0.67 (FCPT: $X^2 = 37.10$, d.f. = 12, $p < 0.001$; table 1). Having excluded LRS records equal to zero, captive-bred fish again displayed significantly lower fitness relative to wild-bred fish (one-tailed permutation test: RRS = 0.81, $p < 0.001$). The average reduction in the mean LRS in the mixed population calculated across the six cohorts, compared with a theoretical population of purely wild-bred fish, was 22.2%. We report the variance in LRS for both provenances across all cohorts in electronic supplementary material, table S4.

The population-level analysis revealed a significant negative relationship between our density-independent population productivity measure and the proportion of captive-bred fish in a spawning cohort (adjusted $R^2 = 0.11$, $F_{1,41} = 6.44$, $p$-value = 0.015; figure 1*b*). Population productivity at the mean value of the proportion captive-bred fish across the 43-year period (0.15) was reduced, on average, by 9.78% (back-transformed from the log scale), relative to a hypothetical pure population (proportion captive-bred fish = 0). For the six cohorts where we had LRS data, the mean captive-bred proportion was 0.19 and the reduction in population productivity was 12.4%. Similar results were found using our alternative 'ova-per-ovum' productivity measure.

## 4. Discussion

Numerous studies have consistently revealed the reduced ability of captive-bred salmonines to survive and breed successfully in the wild [7] (electronic supplementary material, table S1), but few studies have been able to estimate the lifetime contribution of captive-bred fish to subsequent generations relative to wild-bred fish. This information is vital as it quantifies the net fitness impacts of captive-rearing at the individual level, which in turn can inform population-level analyses and eco-evolutionary modelling studies. We demonstrate that captive-bred Atlantic salmon,

predominantly of local origin, that had the opportunity to spawn in the wild exhibited lower LRS than wild-bred conspecifics, that this fitness reduction was apparent for both females and males and that the inferior performance of captive-bred fish depressed overall population productivity. These findings mirror those of the steelhead trout studies [13,14,16]. The steelhead studies were seminal as they provided some of the first multi-generational, pedigree-derived, unbiased RRS estimates between wild-spawning captive- and wild-bred fish. To the best of our knowledge, this has not been shown before for any population of Atlantic salmon. As such, our results have implications for ranching and stocking programmes across the native range of Atlantic salmon, where these practices are often used for either angling gains, mitigation for dam-impounded rivers, or as a conservation strategy.

The significantly lower LRS of captive-bred compared to wild-bred salmon remained after we excluded LRS records equal to zero, consistent with transgenerational effects of the hatchery environment on the survival of offspring produced by captive-bred parents. To further explore this, we used our pedigree to estimate the LRS of wild-bred offspring as a function of whether they had zero, one or two captive-bred parents (electronic supplementary material, figure S5). That is, we compared the mean number of grandchildren produced by captive-bred × captive-bred matings, captive-bred × wild-bred matings and wild-bred × wild-bred matings across the six cohorts. Statistical power was limited but we did find a non-significant trend where wild-bred fish with either one or two captive-bred parents had decreased LRS relative to fish with two wild-bred parents (electronic supplementary material, figure S5). This pattern is consistent with findings in both steelhead [26] and Atlantic salmon [20] that demonstrated transgenerational carry-over effects from the hatchery environment. However, neither Evans *et al.* [20] nor our study were able to disentangle if such transgenerational effects reflect genetic or non-genetic inheritance. Studies on steelhead [13,14,16,26] and brook trout [57] (*Salvelinus fontinalis*, Mitchill, 1814) show that one or two generations of captive-rearing are sufficient to induce maladaptation in captive-bred fish, or their descendants. This may reflect inadvertent domestication selection, relaxed natural and sexual selection, or epigenetic inheritance. While studies of salmonines are beginning to reveal epigenetic effects of hatchery rearing [19,21], further

study is required before generalizations can be made regarding the relative importance of genetically versus epigenetically mediated maladaptation.

In our study, we could only assign parents to offspring that themselves survived to recruit as adults and in the majority of cases (78.68%), only a single parent could be assigned, owing to incomplete sampling of candidate parents. For the minority of cases in which two parents could be assigned, 28.68% involved captive-bred× captive-bred matings, 20.59% involved captive-bred × wild-bred matings and 50.74% involved wild-bred × wild-bred matings. The mean LRS of captive-bred × captive-bred pairs ($n = 39$) was 0.26, for captive-bred × wild-bred pairs ($n = 28$) was 0.64, and for wild-bred × wild-bred pairs was 0.57. These numbers were too low to undertake meaningful statistical analysis, but the pattern is consistent with transgenerational effects of the hatchery environment on offspring survival, with possible non-additive effects of parental provenance.

Previous studies of Burrishoole salmon [10,36] demonstrated that increased captive-born intrusion depressed the freshwater productivity of the overall population. A similar result is known for Scottish salmon [58]. Our study goes further, using recruits per spawner as a measure of productivity. Crucially, this productivity measure incorporates the marine life stage (lacking in [10] and [36]), which accounts for potential provenance-specific variation in marine survival. This facilitated meaningful comparison of fitness across the entire life cycle. Given the higher fecundity of captive- relative to wild-bred fish (Marine Institute 1970–2012, unpublished data), it might be expected that larger proportional intrusion from captive-bred fish would increase the productivity of the mixed population. However, as this study (as well as [10] and [36]) demonstrated, the opposite response was observed. This was corroborated by our pedigree-derived, individual-level, LRS data: we estimated a reduction in mean LRS of 22.2% across the six cohorts for the mixed population relative to a hypothetical pure wild-bred population. One potential explanation is that in years with more captive-bred fish—which are more fecund than wild-bred fish—there are more initial fry in total and hence there is stronger competition among offspring for feeding territories and, hence, lower juvenile survival for both provenances. However, our population-level analysis of productivity accounted for density dependence [59] and still found an effect of captive-bred intrusion, which implies that either a higher fraction of captive-bred fish fail to spawn successfully, or their offspring survive less well relative to the offspring of wild-bred parents [53]. Tentative evidence for the latter explanation was provided by our additional analysis where LRS records of zero were excluded, and the analysis of grand-offspring numbers presented in electronic supplementary material, figure S5. Another potential route for the intrusion of captive-bred genes into the wild population is via a higher tolerance by captive-bred females to matings with subdominant males [60].

Even if offspring of captive-bred fish are initially competitively superior to offspring of wild-bred fish (as has been found for wild-bred offspring of farmed salmon [61]), this advantage is more than outweighed by processes that reduce their overall survival. For example, captive-bred females produce smaller eggs than wild-bred females, potentially due to relaxed selection [60] that may be associated with a correlated increase in egg number. In the wild, fry emerging from smaller eggs are likely to suffer higher early mortality [62,63], and hence this could contribute to the overall lower LRS of captive-bred fish. McGinnity et al. [10] further speculated that various bio-energetic and phenological mechanisms (e.g. winter energy use and timing of fry swim-up) could lead to the offspring of captive-bred fish having lower freshwater survival than offspring of wild-bred fish. Additionally, the offspring of captive-bred fish may perform less well during the smolt/oceanic life stage, again, reducing population productivity. As stated earlier, a potential source of bias in our data stems from the fact that captive-bred fish were predominantly sampled as upstream migrants, whereas wild-bred were predominantly sampled as downstream migrants. While this may have impacted our findings quantitatively, we believe our overall results to be robust in qualitative terms to this potential source of bias (electronic supplementary material, text S1).

In conclusion, our results bolster the consensus that captive-bred animals often have lower fitness in wild environments than wild-bred conspecifics and their interbreeding can depress the productivity of the recipient populations. This raises questions regarding whether supplementation represents a viable mitigation strategy. McGinnity et al. [10] found that, under projected future climate regimes, high levels of hatchery influence have the potential to depress productivity to an extent that threatens population persistence. Moreover, reductions in population productivity may be accompanied by concomitant reductions in effective population size and the loss of adaptive traits [64] which negatively impacts long-term evolutionary potential. Considering this, and given the scale with which Atlantic salmon are subjected to stocking and ranching across their range, there is the potential for wide-scale population declines if stocking and ranching continue without due consideration to what causes captive-bred fish or their descendants to perform poorly relative to wild-bred fish.

Ethics. All fish sampling was done under permission from Rialtas na hÉireann.

Data accessibility. LRS and pedigree data are archived at https://doi.org/10.20393/1b6fed63-4d4b-40f5-9473-32e8210e605a. The code used to analyse LRS and pedigree data, data and code used in estimating unbiased measures of RRS, and data and code related to productivity analyses are archived at http://data.marine.ie/geonetwork/srv/eng/catalog.search#/metadata/ie.marine.data:dataset.4346.

Authors' contributions. R.J.O., T.E.R., P.M., and E.d.E. conceptualized the paper. R.J.O. and T.E.R. designed the LRS and productivity analyses, with R.J.O. conducting the analyses P.M., C.R.P., T.A., S.E.J., R.P. and P.A.P. conceived the original Burrishoole pedigree construction project. P.M., G.R. and R.P. facilitated data collection and P.M., G.R., R.P. and E.d.E. provided access to historical datasets. T.A., S.E.J. and C.R.P. generated the molecular data and constructed the pedigree. R.J.O. and T.E.R. wrote the first draft of the manuscript, with all co-authors contributing to subsequent drafts.

Competing interests. The authors declare they have no conflicting interests with the work herein.

Funding. R.J.O., T.E.R., P.M. and P.A.P. were supported by Science Foundation Ireland, the Marine Institute and the Department for the Economy, Northern Ireland, under the Investigators Programme Grant Number SFI/15/IA/3028. P.M. also received support from a Beaufort Marine Research Award in Fish Population Genetics from Rialtas na hÉireann. TER was funded by an ERC Starting Grant (639192) and an SFI ERC Support Award. R.P., G.R., E.d.E. and the long-term salmon sampling and data collection were funded by the Marine Institute, Ireland. T.A., S.E.J. and C.R.P. were funded by the Academy of Finland (grant numbers 141231, 137710, 307593, 302873 and 31939).

**Acknowledgements.** The authors sincerely thank the many staff of the Marine Institute research station in Burrishoole who have maintained the migratory fish census over six decades, especially Deirdre Cotter as manager of the Burrishoole Atlantic salmon ranching programme. The authors would like to thank Prof. Gary Carvalho and three anonymous referees whose comments greatly improved both the clarity and quality of the presented work. R.J.O. would like to thank Ann Barber and Saoirse McMahon for their friendship during a difficult time in the author's life. T.A., S.E.J. and C.R.P. would like to thank Katja Salminen and a number of short-term trainees for skilful laboratory assistance.

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
