## [Reviewer comments · Proceedings of the Royal Society B: Biological Sciences]

Review History

RSPB-2020-1671.R0 (Original submission)

Review form: Reviewer 1

Recommendation

Major revision is needed (please make suggestions in comments)

Scientific importance: Is the manuscript an original and important contribution to its field?

Acceptable

General interest: Is the paper of sufficient general interest?

Acceptable

Quality of the paper: Is the overall quality of the paper suitable?

Marginal

Is the length of the paper justified?

No

Should the paper be seen by a specialist statistical reviewer?

Yes

Do you have any concerns about statistical analyses in this paper? If so, please specify them explicitly in your report.

Yes

It is a condition of publication that authors make their supporting data, code and materials available - either as supplementary material or hosted in an external repository. Please rate, if applicable, the supporting data on the following criteria.

Is it accessible?

Yes

Is it clear?

Yes

Is it adequate?

Yes

Do you have any ethical concerns with this paper?

No

Comments to the Author

The release of captive-bred individuals into natural environments is broadly conducted for sustaining wild animal populations, though the negative aspects of the procedures are reported by many researchers. In that context, O'sullivan et al. demonstrated the reduction of captive-bred salmon's fitness in the natural environments. I think the main results are clear, but the interpretation of these results may be difficult because anadromous salmonid species including Atlantic salmon have very complex reproductive systems. However, the authors did not consider this point throughout the manuscript.

This study conducted the population level analysis, and anadromous population of salmonid species like the Burrishoole Atlantic salmon contain both anadromous and river resident individuals, i.e., two types of life history tactics. Reproductive systems are different between the two types. In male, anadromous individuals are large and try to monopolize female, while river resident individuals are relatively small and show sneaking behaviour at releasing the sperm. I do not know in the case of the studied population, anadromous and river resident individuals sometimes appear in female within same salmonid population. They are also different in body size, i.e., fecundity is different.

Therefore, the reproductive success at one spawning occasion is very different between the two types, though it is open to discussion which is more adaptive when considering lifetime reproductive success (LRS). So, I recommend the authors to reanalyze the data with considering the difference in life history tactics in addition to sex and wild/captive bred.

Minor comments:

Line 90-98: The negative effects of released individuals appear on not only wild conspecifics but also other sympatric species and/or local communities. In particular, these effects are well studied in salmonid species. Please introduce this point in somewhere in Introduction.

Line 124-126: Please introduce counterexamples of these statements. For example, some studies revealed that hatchery fish outcompete wild fish, and they gain higher growth and survival than wild fish. Throughout the manuscript, the lack of introducing counterexamples may lead the readers misunderstanding the scientific backgrounds.

Methods: Is Atlantic salmon iteroparous or heteroparous? Generally, anadromous populations of salmonid species show complex reproductive strategies as I described above. This point must be very important to consider individual's fitness, but the authors didn't do it.

Line 439-440: Please explain this part more carefully. If the competitive dominance is determined by body size, off-spring of captive-bred fish become subordinate because body size of offspring is egg size dependent, and captive-bred fish have smaller eggs than wild fish. Even if offsprings of captive-bred fish are smaller than that of wild fish, however, they may become dominant by being vastly outnumbered because fecundity of captive-bred fish is larger than wild fish.

Review form: Reviewer 2

Recommendation

Major revision is needed (please make suggestions in comments)

Scientific importance: Is the manuscript an original and important contribution to its field?

Good

General interest: Is the paper of sufficient general interest?

Good

Quality of the paper: Is the overall quality of the paper suitable?

Good

Is the length of the paper justified?

Yes

Should the paper be seen by a specialist statistical reviewer?

No

Do you have any concerns about statistical analyses in this paper? If so, please specify them explicitly in your report.

Yes

It is a condition of publication that authors make their supporting data, code and materials available - either as supplementary material or hosted in an external repository. Please rate, if applicable, the supporting data on the following criteria.

Is it accessible?

Yes

Is it clear?

Yes

Is it adequate?

Yes

Do you have any ethical concerns with this paper?

No

Comments to the Author

Major Comments

Love this paper, great work and very important to conservation and fundamental biology. Has

been a pleasure to review!

My best advice is to lean more into the use of quantitative data to inform hatchery practices. The findings are novel but they corroborate what is already very well known in the literature about hatcheries. At the risk of being the reviewer that tries to tell the authors what they should do, I really was hoping to see the quantitative data used to estimate thresholds in the wild salmon population where the hatchery could have positive effects (ie. Population so incredibly low that there is no alternative option) and where it becomes negative. This should be a relatively simple model or simulation to make a suggestion about when a hatchery should be used and this would make a huge and lasting impact. Perhaps this is already in the works separately, but given the data presented here and the story it fits so well to provide some empirical advice to scientists and managers about how to use the findings to advance their conservation efforts. As someone that works closely with Atlantic salmon in rivers where hatcheries are often used to produce smolts, it is known that the hatchery fish have poorer performance and that their genes are slowly degrading the wild population but it is not known at what point it is important to shut it down in favour of wild spawning. In fact this is a topic of intense debate with no steering mechanism to help guide. Using this manuscript I could go to the agency and say ok, the hatchery fish are about 30% as successful as the wild fish in spawning, and they would argue ok well we can triple the number of smolts we release. Which is of course not the point. I don't expect a rewrite or a huge shift in the manuscript, but I hope the authors can use this comment to add some value to the manuscript in terms of how the results will be applied in a very important and often politically challenging conservation landscape.

Surprised not to see any of the work by Charles Perrier and Louis Bernatchez referred to anywhere, as they have produced some important literature on domestication of Atlantic salmon (I am not affiliated with either of these people in any way). Perrier et al. have also shown that a few generations of domestication greatly affects the fitness potential of Atlantic salmon and the findings would make an excellent contrast

The introduction reads very well but should be cut down and more explicitly focused on the mechanisms of lifetime reproductive success and how the experiments are to be designed to develop new evidence about hatchery introgression

Minor comments

L128- could this be clarified by stating that the phenotypes can change due to reaction norms and the phenotypic frequencies should also shift because natural selection is not removing the relatively unfit ones from the population

L133- suggest end this paragraph with a question related to lifetime reproductive success and the mechanisms driving this phenomenon, which will set up the experiments in this study

L144- clarify that the genetic changes are the result of processes of artificial selection

L148- should not begin the paragraph with "therefore" - requires an opening statement

L156- much of this diverges away from the main thesis of the article and I suggest it would make very good discussion material. Introduction should be more focused specifically on what lifetime reproductive success is and how it links individual to population level process, how it is realized through migration and spawning, and how it is threatened by maladaptation caused by hatcheries. Could also refer to some work by Hendry and Beall (I am not affiliated with either of these people) on the spawning behaviour of Atlantic salmon

L163-169- this is nice information but I suggest to cut it from the intro (maybe move to discussion) and just start by introducing the design at this point

L196- what is microtagging

L208- does this mean 100% of spawners are noted?

L208- what is the contribution of mature male parr and how might this affect the parentage assignments?

L229- unclear why this assumption is necessary if you know the identity of all the spawners?

L234- what is productivity in this sense? The ratio of recruits to spawners?

L239- how is this calculated? Fixed number of eggs/kg Females?

L242- fitness should be addressed in the introduction more explicitly

L250- must be more clear about how this was accounted for, briefly summarise rather than must referring to another study

L252- unclear what is being permuted here.. in each year each individual is assigned a relative reproductive success based on the number of its children that return to spawn.. I don't see a Table S2 so I assume it is Table 2

Year	Individual ID	Provenance	RRS
1977	001	Wild	1.0
1977	002	Hatchery	0.3
1977	003	hatchery	0.2

Is this how the data look? A RRS for each individual for many years?

Table 2- unclear why these are not separated for wild and hatchery?

Table 2- why are there some RRS values >1? Wild fish are assigned a value of 1, so I can't figure out how values could exceed this

L260- the reasoning behind the permutation tests and Fishers test are not well explained to me here

L282- this is a clever idea to exclude non-spawners, I like it

L319- number of knots allowed in the smoother could make a large difference in the GAM, was this specified in any way?

L320- really interesting idea, why the residuals act as a density-corrected productivity metric is not entirely obvious

Figure S1- not including the supplementary figures in this document makes the review quite challenging

L409- is there potential bias in the ability to identify the origin of hatchery compared to wild parents?

L430- lots of density dependence literature for juvenile salmonids that could be referred to here- overall this is a great paragraph, very nicely explained!

L438- the potential contribution of mature male parr to the population is glaringly absent throughout- articles such as Saura et al. indicate that mature parr can have a substantial influence on parentage in populations (Hutchings and Myers show this as well I think)... worth mention in the introduction, methods (how underrepresented are these?), and perhaps even at length in the discussion now. Perhaps there is good reasoning for not having more representation, but others have done so (e.g. Richard et al. 2003 Molecular Ecology)

L455- I read this as saying that wild fish were only sampled as kelts moving out after spawning, but we know there can be (and often is) lots of mortality after spawning, so I would expect that this would have a huge potential bias (with lots of variance among years)... wild fish are clearly less likely to be represented in the sample and offspring from wild parents are less likely to have an assigned parentage (if I understand correctly).. it is quite crucial to explain in more detail why this is not a problem

L461- having quantitative data on this should lend well to a simulation study- hatcheries seem to work only ok in the most dire of circumstances, when the populations are so low that they could never otherwise recover and need rescue by hatcheries. Could a quick simulation be conducted to determine at what population threshold of wild salmon should trigger the start of a hatchery? It should be relatively simple to estimate at what point hatcheries have diminishing returns and damaging effects on the population

Review form: Reviewer 3

Recommendation

Accept with minor revision (please list in comments)

Scientific importance: Is the manuscript an original and important contribution to its field?

Excellent

General interest: Is the paper of sufficient general interest?

Good

Quality of the paper: Is the overall quality of the paper suitable?

Good

Is the length of the paper justified?

Yes

Should the paper be seen by a specialist statistical reviewer?

No

Do you have any concerns about statistical analyses in this paper? If so, please specify them explicitly in your report.

No

It is a condition of publication that authors make their supporting data, code and materials available - either as supplementary material or hosted in an external repository. Please rate, if applicable, the supporting data on the following criteria.

Is it accessible?

Yes

Is it clear?

Yes

Is it adequate?

Yes

Do you have any ethical concerns with this paper?

No

Comments to the Author

The manuscript describes one of few studies have been able to estimate the lifetime contribution of captive-bred fish to subsequent generations relative to wild-bred fish. A key finding is evidence of transgenerational carry-over effects from the hatchery environment that affect the fitness of wild-bred offspring in Atlantic salmon. In my view, this represents an important study of broad interest to researchers whether their interests lie in salmonid fish ecology, the consequences of farm:wild animal interactions, population-restoration efforts, or conservation biology per se. It also represents an excellent example of a long-term study whose results have been helpfully and impressively informed by molecular genetic techniques.

I have one 'major' point and a few minor comments for the authors' consideration.

MAJOR: Lines 213-214: The text reads "94% of wild-bred fish were tissue sampled as kelts in the downstream traps, while 95% of captive-bred fish were sampled pre-spawning in the upstream traps."

So, the wild-bred fish were captured after spawning had occurred while the captive-bred fish were captured before spawning had occurred. I may have missed this information, but how many days prior to and following spawning were the captive-bred and wild-bred fish, respectively, likely to have been captured?

Given life-history theory, *ceteris paribus* it is possible that the wild-bred fish were more likely to experience costs of reproduction and die prior to downstream migration as kelts. This would

mean that the comparison of LRS between wild-bred and captive-bred salmon might include only the hardiest of the post-spawning wild-bred salmon, whereas the upstream trap captured all captive-bred fish.

Given that this could favour any comparison of LRS towards wild-bred salmon (a key finding of the present ms), I feel that this potential caveat merits consideration and mention by the authors.

MINOR:

1. Line 69 and elsewhere: The authors report differences in average LRS, but I wonder if they could not also report variation in LRS? Among other things, estimates of individual variability in LRS could inform estimates of effective population size, N_e .
2. Line 105: "...the life cycle stage...". For non-salmon scientists, it would be helpful to mention the age at smoltification (i.e., seaward migration) relative to the age at maturity (i.e., the age at return from the sea).
3. Line 105: Replace "where" with "at which".
4. Line 116: Egg-to-smolt survival rates of 10% would be high, according to the data presented by Hutchings & Jones (1998. CJFAS). I think that their average estimate is less than 2%.
5. Lines 291 and 297: There is no such entity as 'population fitness'. Fitness can only be defensibly applied to individuals. Please revise.
6. Lines 316-317: I like this data-fitting approach.

Decision letter (RSPB-2020-1671.R0)

25-Aug-2020

Dear Mr O'Sullivan:

Your manuscript has now been peer reviewed and the reviews have been assessed by an Associate Editor. The reviewers' comments (not including confidential comments to the Editor) and the comments from the Associate Editor are included at the end of this email for your reference. As you will see, the reviewers and the Editors have raised some concerns with your manuscript and we would like to invite you to revise your manuscript to address them.

When submitting your revision please upload a file under "Response to Referees" - in the "File Upload" section. This should document, point by point, how you have responded to the reviewers' and Editors' comments, and the adjustments you have made to the manuscript. We

require a copy of the manuscript with revisions made since the previous version marked as 'tracked changes' to be included in the 'response to referees' document.

Research ethics:

Use of animals and field studies:

It is a condition of publication that you make available the data and research materials supporting the results in the article. Please see our Data Sharing Policies (<https://royalsociety.org/journals/authors/author-guidelines/#data>). Datasets should be deposited in an appropriate publicly available repository and details of the associated accession number, link or DOI to the datasets must be included in the Data Accessibility section of the article (<https://royalsociety.org/journals/ethics-policies/data-sharing-mining/>). Reference(s) to datasets should also be included in the reference list of the article with DOIs (where available).

If you wish to submit your data to Dryad (<http://datadryad.org/>) and have not already done so you can submit your data via this link [http://datadryad.org/submit?journalID=RSPB&manu=\(Document not available\)](http://datadryad.org/submit?journalID=RSPB&manu=(Document%20not%20available)), which will take you to your unique entry in the Dryad repository.

Online supplementary material will also carry the title and description provided during submission, so please ensure these are accurate and informative. Note that the Royal Society will not edit or typeset supplementary material and it will be hosted as provided. Please ensure that

the supplementary material includes the paper details (authors, title, journal name, article DOI). Your article DOI will be 10.1098/rspb.[paper ID in form xxxx.xxxx e.g. 10.1098/rspb.2016.0049].

Please submit a copy of your revised paper within three weeks. If we do not hear from you within this time your manuscript will be rejected. If you are unable to meet this deadline please let us know as soon as possible, as we may be able to grant a short extension.

Best wishes,
Professor Gary Carvalho
mailto: proceedingsb@royalsociety.org

Associate Editor
Board Member: 1

Comments to Author:

The reviewers are broadly supportive the work but indicate several important revisions / clarifications to the work. Please take care to address their concerns thoroughly, including clear rebuttals where relevant.

Reviewer(s)' Comments to Author:

Referee: 1

Comments to the Author(s)

The release of captive-bred individuals into natural environments is broadly conducted for sustaining wild animal populations, though the negative aspects of the procedures are reported by many researchers. In that context, O'sullivan et al. demonstrated the reduction of captive-bred salmon's fitness in the natural environments. I think the main results are clear, but the interpretation of these results may be difficult because anadromous salmonid species including Atlantic salmon have very complex reproductive systems. However, the authors did not consider this point throughout the manuscript.

This study conducted the population level analysis, and anadromous population of salmonid species like the Burrishoole Atlantic salmon contain both anadromous and river resident individuals, i.e., two types of life history tactics. Reproductive systems are different between the two types. In male, anadromous individuals are large and try to monopolize female, while river resident individuals are relatively small and show sneaking behaviour at releasing the sperm. I do not know in the case of the studied population, anadromous and river resident individuals sometimes appear in female within same salmonid population. They are also different in body size, i.e., fecundity is different.

Therefore, the reproductive success at one spawning occasion is very different between the two types, though it is open to discussion which is more adaptive when considering lifetime reproductive success (LRS). So, I recommend the authors to reanalyze the data with considering the difference in life history tactics in addition to sex and wild/captive bred.

Minor comments:

Line 90-98: The negative effects of released individuals appear on not only wild conspecifics but also other sympatric species and/or local communities. In particular, these effects are well studied in salmonid species. Please introduce this point in somewhere in Introduction.

Line 124-126: Please introduce counterexamples of these statements. For example, some studies revealed that hatchery fish outcompete wild fish, and they gain higher growth and survival than

wild fish. Throughout the manuscript, the lack of introducing counterexamples may lead the readers misunderstanding the scientific backgrounds.

Methods: Is Atlantic salmon iteroparous or heteroparous? Generally, anadromous populations of salmonid species show complex reproductive strategies as I described above. This point must be very important to consider individual's fitness, but the authors didn't do it.

Line 439-440: Please explain this part more carefully. If the competitive dominance is determined by body size, off-spring of captive-bred fish become subordinate because body size of offspring is egg size dependent, and captive-bred fish have smaller eggs than wild fish. Even if offsprings of captive-bred fish are smaller than that of wild fish, however, they may become dominant by being vastly outnumbered because fecundity of captive-bred fish is larger than wild fish.

Referee: 2

Comments to the Author(s)

Major Comments

Love this paper, great work and very important to conservation and fundamental biology. Has been a pleasure to review!

My best advice is to lean more into the use of quantitative data to inform hatchery practices. The findings are novel but they corroborate what is already very well known in the literature about hatcheries. At the risk of being the reviewer that tries to tell the authors what they should do, I really was hoping to see the quantitative data used to estimate thresholds in the wild salmon population where the hatchery could have positive effects (ie. Population so incredibly low that there is no alternative option) and where it becomes negative. This should be a relatively simple model or simulation to make a suggestion about when a hatchery should be used and this would make a huge and lasting impact. Perhaps this is already in the works separately, but given the data presented here and the story it fits so well to provide some empirical advice to scientists and managers about how to use the findings to advance their conservation efforts. As someone that works closely with Atlantic salmon in rivers where hatcheries are often used to produce smolts, it is known that the hatchery fish have poorer performance and that their genes are slowly degrading the wild population but it is not known at what point it is important to shut it down in favour of wild spawning. In fact this is a topic of intense debate with no steering mechanism to help guide. Using this manuscript I could go to the agency and say ok, the hatchery fish are about 30% as successful as the wild fish in spawning, and they would argue ok well we can triple the number of smolts we release. Which is of course not the point. I don't expect a rewrite or a huge shift in the manuscript, but I hope the authors can use this comment to add some value to the manuscript in terms of how the results will be applied in a very important and often politically challenging conservation landscape.

Surprised not to see any of the work by Charles Perrier and Louis Bernatchez referred to anywhere, as they have produced some important literature on domestication of Atlantic salmon (I am not affiliated with either of these people in any way). Perrier et al. have also shown that a few generations of domestication greatly affects the fitness potential of Atlantic salmon and the findings would make an excellent contrast

The introduction reads very well but should be cut down and more explicitly focused on the mechanisms of lifetime reproductive success and how the experiments are to be designed to develop new evidence about hatchery introgression

Minor comments

L128- could this be clarified by stating that the phenotypes can change due to reaction norms and the phenotypic frequencies should also shift because natural selection is not removing the relatively unfit ones from the population

L133- suggest end this paragraph with a question related to lifetime reproductive success and the mechanisms driving this phenomenon, which will set up the experiments in this study

L144- clarify that the genetic changes are the result of processes of artificial selection

L148- should not begin the paragraph with “therefore” - requires an opening statement

L156- much of this diverges away from the main thesis of the article and I suggest it would make very good discussion material. Introduction should be more focused specifically on what lifetime reproductive success is and how it links individual to population level process, how it is realized through migration and spawning, and how it is threatened by maladaptation caused by hatcheries. Could also refer to some work by Hendry and Beall (I am not affiliated with either of these people) on the spawning behaviour of Atlantic salmon

L163-169- this is nice information but I suggest to cut it from the intro (maybe move to discussion) and just start by introducing the design at this point

L196- what is microtagging

L208- does this mean 100% of spawners are noted?

L208- what is the contribution of mature male parr and how might this affect the parentage assignments?

L229- unclear why this assumption is necessary if you know the identity of all the spawners?

L234- what is productivity in this sense? The ratio of recruits to spawners?

L239- how is this calculated? Fixed number of eggs/kg Females?

L242- fitness should be addressed in the introduction more explicitly

L250- must be more clear about how this was accounted for, briefly summarise rather than must referring to another study

L252- unclear what is being permuted here.. in each year each individual is assigned a relative reproductive success based on the number of its children that return to spawn.. I don't see a Table S2 so I assume it is Table 2

Year | Individual ID | Provenance | RRS

1977 | 001 | Wild | 1.0

1977 | 002 | Hatchery | 0.3

1977 | 003 | hatchery | 0.2

Is this how the data look? A RRS for each individual for many years?

Table 2- unclear why these are not separated for wild and hatchery?

Table 2- why are there some RRS values >1? Wild fish are assigned a value of 1, so I can't figure out how values could exceed this

L260- the reasoning behind the permutation tests and Fishers test are not well explained to me here

L282- this is a clever idea to exclude non-spawners, I like it

L319- number of knots allowed in the smoother could make a large difference in the GAM, was this specified in any way?

L320- really interesting idea, why the residuals act as a density-corrected productivity metric is not entirely obvious

Figure S1- not including the supplementary figures in this document makes the review quite challenging

L409- is there potential bias in the ability to identify the origin of hatchery compared to wild parents?

L430- lots of density dependence literature for juvenile salmonids that could be referred to here- overall this is a great paragraph, very nicely explained!

L438- the potential contribution of mature male parr to the population is glaringly absent throughout- articles such as Saura et al. indicate that mature parr can have a substantial influence on parentage in populations (Hutchings and Myers show this as well I think)... worth mention in the introduction, methods (how underrepresented are these?), and perhaps even at length in the discussion now. Perhaps there is good reasoning for not having more representation, but others have done so (e.g. Richard et al. 2003 Molecular Ecology)

L455- I read this as saying that wild fish were only sampled as kelts moving out after spawning, but we know there can be (and often is) lots of mortality after spawning, so I would expect that this would have a huge potential bias (with lots of variance among years)... wild fish are clearly less likely to be represented in the sample and offspring from wild parents are less likely to have

an assigned parentage (if I understand correctly).. it is quite crucial to explain in more detail why this is not a problem

L461- having quantitative data on this should lend well to a simulation study- hatcheries seem to work only ok in the most dire of circumstances, when the populations are so low that they could never otherwise recover and need rescue by hatcheries. Could a quick simulation be conducted to determine at what population threshold of wild salmon should trigger the start of a hatchery? It should be relatively simple to estimate at what point hatcheries have diminishing returns and damaging effects on the population

Referee: 3

Comments to the Author(s)

The manuscript describes one of few studies have been able to estimate the lifetime contribution of captive-bred fish to subsequent generations relative to wild-bred fish. A key finding is evidence of transgenerational carry-over effects from the hatchery environment that affect the fitness of wild-bred offspring in Atlantic salmon. In my view, this represents an important study of broad interest to researchers whether their interests lie in salmonid fish ecology, the consequences of farm:wild animal interactions, population-restoration efforts, or conservation biology per se. It also represents an excellent example of a long-term study whose results have been helpfully and impressively informed by molecular genetic techniques.

I have one 'major' point and a few minor comments for the authors' consideration.

MAJOR: Lines 213-214: The text reads "94% of wild-bred fish were tissue sampled as kelts in the downstream traps, while 95% of captive-bred fish were sampled pre-spawning in the upstream traps."

So, the wild-bred fish were captured after spawning had occurred while the captive-bred fish were captured before spawning had occurred. I may have missed this information, but how many days prior to and following spawning were the captive-bred and wild-bred fish, respectively, likely to have been captured?

Given life-history theory, *ceteris paribus* it is possible that the wild-bred fish were more likely to experience costs of reproduction and die prior to downstream migration as kelts. This would mean that the comparison of LRS between wild-bred and captive-bred salmon might include only the hardiest of the post-spawning wild-bred salmon, whereas the upstream trap captured all captive-bred fish.

Given that this could favour any comparison of LRS towards wild-bred salmon (a key finding of the present ms), I feel that this potential caveat merits consideration and mention by the authors.

MINOR:

1. Line 69 and elsewhere: The authors report differences in average LRS, but I wonder if they could not also report variation in LRS? Among other things, estimates of individual variability in LRS could inform estimates of effective population size, N_e .

2. Line 105: "...the life cycle stage...". For non-salmon scientists, it would be helpful to mention the age at smoltification (i.e., seaward migration) relative to the age at maturity (i.e., the age at return from the sea).

3. Line 105: Replace "where" with "at which".

4. Line 116: Egg-to-smolt survival rates of 10% would be high, according to the data presented by Hutchings & Jones (1998. CJFAS). I think that their average estimate is less than 2%.

5. Lines 291 and 297: There is no such entity as 'population fitness'. Fitness can only be defensibly applied to individuals. Please revise.
6. Lines 316-317: I like this data-fitting approach.

Author's Response to Decision Letter for (RSPB-2020-1671.R0)

See Appendix A.

Decision letter (RSPB-2020-1671.R1)

28-Sep-2020

Dear Mr O'Sullivan

I am pleased to inform you that your manuscript entitled "Captive-bred Atlantic salmon released into the wild have fewer offspring than wild-bred fish and decrease population productivity." has been accepted for publication in Proceedings B.

Open Access

You are invited to opt for Open Access, making your freely available to all as soon as it is ready for publication under a CCBY licence. Our article processing charge for Open Access is £1700. Corresponding authors from member institutions (<http://royalsocietypublishing.org/site/librarians/allmembers.xhtml>) receive a 25% discount to these charges. For more information please visit <http://royalsocietypublishing.org/open-access>.

Paper charges

Sincerely,
Professor Gary Carvalho
Editor, Proceedings B
mailto: proceedingsb@royalsociety.org

Associate Editor:
Board Member
Comments to Author:
(There are no comments.)

Appendix A

Response to referees for O’Sullivan et al. - Manuscript ID RSPB-2020-1671

We would like to take this opportunity to thank Prof. Carvalho for his handling of our manuscript, and to thank the three referees, whose helpful and insightful comments have greatly improved the quality and clarity of this research.

This document contains our responses to the referees’ comments, followed by a revised version of the manuscript that highlights where changes have been made. Referee comments are presented below in blue, italicised text with our responses directly below each comment.

Responses to Referee 1-

“The release of captive-bred individuals into natural environments is broadly conducted for sustaining wild animal populations, though the negative aspects of the procedures are reported by many researchers. In that context, O’Sullivan et al. demonstrated the reduction of captive-bred salmon’s fitness in the natural environments. I think the main results are clear, but the interpretation of these results may be difficult because anadromous salmonid species including Atlantic salmon have very complex reproductive systems. However, the authors did not consider this point throughout the manuscript.”

We thank Referee 1 for their comment that “the main results are clear”. The reviewer is correct that Atlantic salmon populations generally display great variation in lifecycle across their distribution. Nonetheless, the lifecycle of salmon in the Burrishoole catchment is comparatively simple, with around 90% of wild-bred fish adhering to a ‘2+ freshwater – grilse’ lifecycle (grilse referring to a fish that has spent one winter feeding at sea; note that captive-bred fish are reared in a hatchery prior to their release into the ocean and, therefore, do not exhibit a wild-living freshwater stage). The simple lifecycle of Burrishoole Atlantic salmon is one of the strengths of our study system as it allows for ease of biological interpretation.

More so, the lifetime reproductive success of fish with infrequent age structure (either going to sea at 1+ or 3+ ages and returning to spawn as multi-sea winter, ‘MSW’, fish) was estimated via the pedigree in the same way as the other fish, and their LRS was

indexed to the appropriate years. This accounted for the variation in lifecycle displayed by Burrishoole Atlantic salmon (see lines 280-693 in Code text file). Similarly, fish that were recorded as spawning across multiple years had their year-specific reproductive success assigned to the appropriate spawning event, rendering our comparisons of both cohort-specific and overall relative reproductive success appropriate and accurate (see lines 280-693 in Code text file).

We do appreciate, however, that this might not have been completely clear to the referee. We have hence revised the manuscript to include more details of this treatment of lifecycle complexity so as to improve the clarity of the methods and any resulting biological interpretation. See lines 182-185 of the revised manuscript for the following revision, *“Fish that survive spawning return to the sea early in the following calendar year. Individual fish in our data that did not display the conventional 2+-grilse lifecycle had their LRS appropriately indexed to the correct years across which they spawned (see lines 280-693 in Code text file available at doi:).”*

“This study conducted the population level analysis, and anadromous population of salmonid species like the Burrishoole Atlantic salmon contain both anadromous and river resident individuals, i.e., two types of life history tactics. Reproductive systems are different between the two types. In male, anadromous individuals are large and try to monopolize female, while river resident individuals are relatively small and show sneaking behaviour at releasing the sperm. I do not know in the case of the studied population, anadromous and river resident individuals sometimes appear in female within same salmonid population. They are also different in body size, i.e., fecundity is different.”

Referee 1 raises an important point regarding our consideration of resident versus anadromous life history tactics. This is a point re-iterated by Referee 2 so we address both referees' concerns here. In many populations, including Burrishoole, some males indeed mature as parr without having first undergone an oceanic feeding migration. There are no resident females in the Burrishoole. The data used in our estimates of lifetime reproductive success were collected from fish traps located at the point of sea entry in the Burrishoole catchment. Therefore, our data include only anadromous

individuals; we did not also sample mature male parr in fresh water during the spawning period. We do not believe that the lack of direct sampling of precocial males in our study biases our results, as McGinnity et al. 2007 found no evidence that males from wild and captive genetic backgrounds differ in their propensities for precocial maturation, in our population. Being able to account for mature male parr in our calculations of individual LRS would only impact the absolute estimates of mean LRS for each provenance (i.e. how many offspring in total are produced by a given individual over its lifetime, accounting for both precocial and anadromous spawning events in the case of males). Crucially, however, it would not influence our estimates of relative reproductive success of captive- and wild-bred salmon, as we are underestimating their absolute LRS by the same amount. We make this point explicit in the revised manuscript at lines 193-197, *“Captive- and wild-bred salmon in Burrishoole have similar propensities for male precociality⁴⁵ so we do not believe their absence from our data biases our estimates of relative reproductive success, as our estimates of absolute LRS for each provenance are expected to be equally biased by failing to account for mature male parr.”*

“Therefore, the reproductive success at one spawning occasion is very different between the two types, though it is open to discussion which is more adaptive when considering lifetime reproductive success (LRS). So, I recommend the authors to reanalyze the data with considering the difference in life history tactics in addition to sex and wild/captive bred.”

Given the lack of difference in propensity for precociality between wild- and captive-bred male salmon in the Burrishoole system (McGinnity et al. 2007), coupled with the complete absence of mature female parr in our system, we do not believe that this issue in any way biases our inferences with respect to the relative reproductive success of wild versus captive-bred fish, which is the key focus of the study. Moreover, because we did not sample mature male parr – rather only anadromous fish – we have no way of reanalysing the individual-level, pedigree-derived data in a way that accounts for life history tactic.

“Line 90-98: The negative effects of released individuals appear on not only wild conspecifics but also other sympatric species and/or local communities. In particular, these effects are well studied in salmonid species. Please introduce this point in somewhere in Introduction.”

We agree with Referee 1 that we need to acknowledge the negative effects that the release of captive-bred individuals can have on community structure in an ecosystem. We have revised the manuscript accordingly (line 97) with the addition of reference to Bradbury et al. In Press.

“Line 124-126: Please introduce counterexamples of these statements. For example, some studies revealed that hatchery fish outcompete wild fish, and they gain higher growth and survival than wild fish. Throughout the manuscript, the lack of introducing counterexamples may lead the readers misunderstanding the scientific backgrounds.”

Generally, there is no longer any doubt that the deliberate or accidental release of captive-bred salmonids has negative long-term consequences for wild populations, as demonstrated in this study, as well as by several others (e.g. Araki et al. 2007; Araki et al. 2009; see Table S1 for a list of such studies) with only a few exceptional cases that show otherwise (e.g. Hess et al. 2012). We do acknowledge in the paper (lines 430-437) that hatchery-born offspring may exhibit initial advantages in growth or competitiveness, but these potential advantages are outweighed by subsequent survival or reproductive disadvantages such that lifetime fitness is typically lower, as demonstrated in the meta-analysis of Christie et al. 2014.

“Methods: Is Atlantic salmon iteroparous or heteroparous? Generally, anadromous populations of salmonid species show complex reproductive strategies as I described above. This point must be very important to consider individual’s fitness, but the authors didn’t do it.”

We are unsure as to what ‘heteroparous’ means but believe Referee 1 may be referring to the iteroparity – semelparity distinction. Atlantic salmon are iteroparous. As stated in our first two responses to Referee 1’s comments, we did consider the complexity of the

lifecycle and treated the data appropriately. In the case of iteroparity, if a given sampled fish spawned multiple times, we accounted for its reproductive success across all spawning years. We have made our treatment of the lifecycle more explicit in the text at lines 182-185 of the revised manuscript, *“Fish that survive spawning return to the sea early in the following calendar year. Individual fish in our data that did not display the conventional 2+-grilse lifecycle had their LRS appropriately indexed to the correct years across which they spawned (see lines 280-693 in Code text file available at doi:).”*

“Line 439-440: Please explain this part more carefully. If the competitive dominance is determined by body size, off-spring of captive-bred fish become subordinate because body size of offspring is egg size dependent, and captive-bred fish have smaller eggs than wild fish. Even if offsprings of captive-bred fish are smaller than that of wild fish, however, they may become dominant by being vastly outnumbered because fecundity of captive-bred fish is larger than wild fish.”

We apologise if this was confusing in our original manuscript. It is indeed true that captive-bred females in this system produce smaller eggs, and hence their wild-born offspring might be smaller than those of wild-bred females, and hence they may be initially competitively inferior. However, the offspring of captive-bred females or males might exhibit faster growth rates early in life, if unintentionally selected for rapid growth in the hatchery environment, or they may be more bold/aggressive “pound for pound”, if the hatchery environment selects for increased aggression. Thus, it remains possible that these latter effects could have outweighed any initial size disadvantage such that offspring of captive-bred fish are initially competitively superior to offspring of wild-bred fish. This remains speculative though, and our results clearly show that across the full life cycle, the LRS of captive-bred is lower than that of wild-bred, indicating that offspring born to captive parents clearly are less likely to survive to adulthood. We have clarified this now in the revised manuscript on lines 430-437, *“Even if offspring of captive-bred fish are initially competitively superior to offspring of wild-bred fish (as has been found for wild-bred offspring of farmed salmon⁶²), this advantage is more than outweighed by processes that reduce their overall survival. For example, captive-bred females produce smaller eggs than wild-bred females, potentially due to relaxed selection⁶¹ that may be associated with a correlated increase in egg number. In the wild, fry emerging from smaller eggs are likely to*

suffer higher early mortality^{63,64} and hence this could contribute to the overall lower LRS of captive-bred fish.”

With respect to the second part of this comment, , captive-bred females have higher egg numbers than wild-bred females, but this initial numerical advantage does not translate to more offspring surviving to recruit (i.e. higher LRS), but again is outweighed by reduced offspring ‘quality’, i.e. ability to survive to adulthood.

Responses to Referee 2-

“Love this paper, great work and very important to conservation and fundamental biology. Has been a pleasure to review!

My best advice is to lean more into the use of quantitative data to inform hatchery practices. The findings are novel but they corroborate what is already very well known in the literature about hatcheries. At the risk of being the reviewer that tries to tell the authors what they should do, I really was hoping to see the quantitative data used to estimate thresholds in the wild salmon population where the hatchery could have positive effects (ie. Population so incredibly low that there is no alternative option) and where it becomes negative. This should be a relatively simple model or simulation to make a suggestion about when a hatchery should be used and this would make a huge and lasting impact. Perhaps this is already in the works separately, but given the data presented here and the story it fits so well to provide some empirical advice to scientists and managers about how to use the findings to advance their conservation efforts. As someone that works closely with Atlantic salmon in rivers where hatcheries are often used to produce smolts, it is known that the hatchery fish have poorer performance and that their genes are slowly degrading the wild population but it is not known at what point it is important to shut it down in favour of wild spawning. In fact this is a topic of intense debate with no steering mechanism to help guide. Using this manuscript I could go to the agency and say ok, the hatchery fish are about 30% as successful as the wild fish in spawning, and they would argue ok well we can triple the number of smolts we release.

Which is of course not the point. I don't expect a rewrite or a huge shift in the manuscript, but I hope the authors can use this comment to add some value to the manuscript in terms of how the results will be applied in a very important and often politically challenging conservation landscape.

Referee 2 raises the point of building a model again with the comment *"L461- having quantitative data on this should lend well to a simulation study- hatcheries seem to work only ok in the most dire of circumstances, when the populations are so low that they could never otherwise recover and need rescue by hatcheries. Could a quick simulation be conducted to determine at what population threshold of wild salmon should trigger the start of a hatchery? It should be relatively simple to estimate at what point hatcheries have diminishing returns and damaging effects on the population"*.

We thank Referee 2 for their very positive comments on our manuscript in general, and for raising an interesting and cogent point about using quantitative data to estimate thresholds, below which hatchery influence may have a positive impact. Referee 2 suggests building a model/simulation to explore such a scenario. However, we believe that there is no threshold below which the accidental/deliberate release of captive-bred fish is safe. We have come to this conclusion from both a quantitative as well as a philosophical standpoint.

Quantitative standpoint:

We report the results of a linear model where we regressed a density-corrected measure of productivity against the proportion of captive-bred fish in a spawning cohort. We found a significant negative linear relationship, that is to say, the population's productivity decreased as the proportion of captive-bred spawners increased. This implies that the best level of hatchery input is zero, because productivity is maximal at the origin of this graph. We appreciate that by looking at the left-hand side of our revised Figure 1b (Figure 3 in the original submission), Referee 2 may have thought a potential demographic boost existed at low levels of hatchery input due to the variance displayed by the data above the regression line. To explore a bit further whether low levels of hatchery influence may have had a positive impact, we fit a generalized additive model (GAM) as well as a quadratic regression model to the data. Neither of these models fit the data better than the previously reported linear model. Therefore, we see no evidence for

any benefit, no matter how small, from hatchery influence on productivity of the wild population, with no demographic boost at low levels of hatchery input.

The above analyses corrected for density effects on population productivity, and the level of annual hatchery input (proportion captive-bred spawners) varied independently of population density (annual number of total spawners, wild + captive). So the primary analyses reported in our original submission do not lend themselves readily to answering the question posed by Referee 2: namely, is there a threshold population size below which some stocking might be appropriate (and if so, how much stocking)? One possible approach here is to test for a statistical interaction between annual hatchery input and population density on (raw, uncorrected) annual population productivity. If we follow the logic of Reviewer 2 correctly here, we might expect to see a positive demographic effect of hatchery input (more hatchery fish = better for overall population productivity) at low population densities, which then switches to a negative effect at higher population densities. We ran this extra analysis and found no statistical support for an interaction here. We also split the data into low population density years versus high population density years (based on the grand mean population density), and fit a GAM to each block of years (to explore possible nonlinear effects that vary with population density), where uncorrected population productivity (log recruits per spawner) was the response variable and annual proportion captive spawners was the explanatory variable. Again, we saw no evidence for a demographic boost at low levels of hatchery input, at either low or high population densities – rather, there were approximately linear negative effects of hatchery input for both the low- and high-density year blocks. Thus, from a purely demographic perspective, we see no evidence in our particular dataset for demographic benefits of stocking at any population density. Moreover, the above ignores evolutionary considerations: even if there were short-term demographic boosts associated with stocking, these are likely more than outweighed by negative long-term impacts on the genetic integrity of the population and hence its resilience in the face of environmental change.

Philosophical standpoint:

The introduction of hatchery-bred fish into a wild population that are allowed to freely interbreed with wild fish irrevocably changes the nature of the recipient wild population. Population supplementation using hatcheries is most often for the purpose of ensuring 'fish for the rod' for recreational anglers, which from a purely demographic viewpoint might be sustainable so long as little to no hatchery fish are allowed to (or accidentally) escape into the wild population. Providing fish solely for human exploitation is consistent with a utilitarian view of nature, that is, the salmon population exists solely for what humanity is able to extract from it. Each salmonid population across the distribution of the species may be uniquely adapted to its natal catchment, often displaying catchment- or river-specific genetic adaptation. Given hatchery fish are typically genetically divergent (to varying degrees) from wild-bred fish, the breeding of captive-bred salmonids with wild-bred salmonids leads to a loss of the intrinsic genetic identity of that salmon population. Each population (and, by extension, the population's genetic identity) has an intrinsic value and 'right' to exist in its natural, unaltered state. It is often impossible to restore the original genetic profile of salmon populations subjected to hatchery operations, meaning the irreversible loss of a unique unit of biological and genetic diversity. While many might argue that some salmon are better than no salmon (i.e. best to stock when things get really dire), this attitude is dangerous in our view as it ignores the root causes of population declines and is risky from a genetic perspective, hence it goes against the Precautionary Principle. Given the global erosion of biological diversity currently being witnessed in the Anthropocene, it is our belief that any practice that further reduces the remaining diversity is, by definition, deleterious and should be prevented.

Surprised not to see any of the work by Charles Perrier and Louis Bernatchez referred to anywhere, as they have produced some important literature on domestication of Atlantic salmon (I am not affiliated with either of these people in any way). Perrier et al. have also shown that a few generations of domestication greatly affects the fitness potential of Atlantic salmon and the findings would make an excellent contrast

We would like to thank Referee 2 for highlighting the lack of reference to the work of both Charles Perrier and Louis Bernatchez. Given the extensive body of literature on the topic of hatchery supplementation, these references were overlooked. The manuscript has been revised at lines 96,124-125, 129, 454 citing the work of both authors with respect to our research questions.

The introduction reads very well but should be cut down and more explicitly focused on the mechanisms of lifetime reproductive success and how the experiments are to be designed to develop new evidence about hatchery introgression”

We agree with Referee 2 that the Introduction as presented in the original submission of this manuscript is too long. We have reduced the length somewhat from 1087 words to 854, to try to focus more on the issues alluded to here. We appreciate that this is still a rather long Introduction, but we believe it necessary in order to properly explain our thesis and layout our aims. See lines 89-161 for the revised Introduction.

“L128- could this be clarified by stating that the phenotypes can change due to reaction norms and the phenotypic frequencies should also shift because natural selection is not removing the relatively unfit ones from the population”

We think there is a little bit of confusion here, as this sentence is explicitly about the phenotype of given individual, and hence its phenotype diverges from that of wild fish only via plasticity (e.g. underpinned by epigenetic modifications). But of course, at the population level, the phenotype distribution of hatchery fish will diverge from wild via both plasticity and microevolution.

“L133- suggest end this paragraph with a question related to lifetime reproductive success and the mechanisms driving this phenomenon, which will set up the experiments in this study”

Manuscript revised as follows: *“The inferior post-release survival and spawning behaviour of captive-bred fish, coupled with negative demographic consequences, raises two*

questions - does the poor performance of individual captive-bred fish translate into reduced overall fitness for the captive-bred group relative to wild-bred fish and, if so, do spawning captive-bred fish affect population productivity?”. See lines 129-134.

“L144- clarify that the genetic changes are the result of processes of artificial selection”

We aim to be very specific about the distinction between artificial and domestication selection. “Artificial selection” refers to targeted improvement of particular traits, e.g. selection for rapid growth or other production traits in farm/commercial strains of plants and animals. In contrast “domestication selection” refers to inadvertent selection in the hatchery environment that plays out “naturally” (i.e. no human is directly deciding which fish gets to live and breed and which fish does not). Therefore, we have not modified this point.

“L148- should not begin the paragraph with “therefore”- requires an opening statement”

We have brought the sentence *“Therefore, interbreeding between captive- and wild-bred fish entails evolutionary risks for the wild population, as introgression of ‘hatchery alleles’ can negatively affect the fitness of hybrids in the wild, potentially depressing population size or productivity¹⁷”* up into the preceding paragraph, thereby, improving the clarity and flow of this section of the Introduction.

“L156- much of this diverges away from the main thesis of the article and I suggest it would make very good discussion material. Introduction should be more focused specifically on what lifetime reproductive success is and how it links individual to population level process, how it is realized through migration and spawning, and how it is threatened by maladaptation caused by hatcheries. Could also refer to some work by Hendry and Beall (I am not affiliated with either of these people) on the spawning behaviour of Atlantic salmon”

We agree that our discussion of hatchery-reared fish straying into non-natal rivers diverged from the main thesis of the article. We have removed these sentences from the revised manuscript.

“L163-169- this is nice information but I suggest to cut it from the intro (maybe move to discussion) and just start by introducing the design at this point”

We agree with the referee’s suggestion to move the relevant information from the Introduction into the Discussion. We have done so and as suggested, introduce the study design at an earlier point in the revised Introduction. Lines 406-412 in the Discussion now read as follows, *“Previous studies of Burrishoole salmon^{10,36} demonstrated that increased captive-born intrusion depressed the freshwater productivity of the overall population. A similar result is known for Scottish salmon⁵⁸. Our study goes further, using recruits per spawner as a measure of productivity. Crucially, this productivity measure incorporates the marine life stage (lacking in¹⁰ and³⁶), which accounts for potential provenance-specific variation in marine survival. This facilitated meaningful comparison of fitness across the entire lifecycle.”*

“L196- what is microtagging”

Microtagging refers to the procedure of injecting a coded wire tag (a length of magnetized stainless steel wire 0.25 mm in diameter) into the nose of fish. The tagging of the fish is carried out prior to its release into the sea. Detail of this has been added to the manuscript at lines 173-175, *“Microtagging refers to the procedure of injecting a coded wire tag (a length of magnetized stainless steel wire 0.25 mm in diameter) into the nose of fish”.*

“L208- does this mean 100% of spawners are noted?”

The total trapping system allows for sampling of 100% of anadromous potential recruits (i.e. total number of potential spawners returning from the ocean), as no systematic sampling is performed on the spawning grounds within the catchment. We have clarified this point in the revised manuscript as follows; *“This trapping regime has allowed for the collection of annual census data based on total counts, phenotypic and genetic sampling of the potential anadromous wild-spawning population See lines 197-201.*

“L208- what is the contribution of mature male parr and how might this affect the parentage assignments?”

“L438- the potential contribution of mature male parr to the population is glaringly absent throughout- articles such as Saura et al. indicate that mature parr can have a substantial influence on parentage in populations (Hutchings and Myers show this as well I think)... worth mention in the introduction, methods (how underrepresented are these?), and perhaps even at length in the discussion now. Perhaps there is good reasoning for not having more representation, but others have done so (e.g. Richard et al. 2003 Molecular Ecology)”

We addressed this point already in response to Referee 1 but, due to its importance, reiterate it here. In many populations, including Burrishoole, some males indeed mature as parr without having first undergone an oceanic feeding migration. There are no resident females in the Burrishoole. The data used in our estimates of lifetime reproductive success were collected from fish traps located at the point of sea entry in the Burrishoole catchment. Therefore, our data include only anadromous individuals; we did not also sample mature male parr in fresh water during the spawning period. We do not believe that the lack of direct sampling of precocial males in our study biases our results, as McGinnity et al. 2007 found no evidence that males from wild and captive genetic backgrounds differ in their propensities for precocial maturation, in our population. Being able to account for mature male parr in our calculations of individual LRS would only impact the absolute estimates of mean LRS for each provenance (i.e. how many offspring in total are produced by a given individual over its lifetime, accounting for both precocial and anadromous spawning events in the case of males). Crucially, however, it would not influence our estimates of relative reproductive success of captive- and wild-bred salmon, as we are underestimating their absolute LRS by the same amount. We make this point explicit in the revised manuscript at lines 193-197, *“Captive- and wild-bred salmon in Burrishoole have similar propensities for male precociality⁴⁵ so we do not believe their absence from our data biases our estimates of relative reproductive success, as our estimates of absolute LRS for each provenance are expected to be equally biased by failing to account for mature male parr.”*

“L229- unclear why this assumption is necessary if you know the identity of all the spawners?”

This assumption is necessary because, while we know the total number of fish that smolted and returned over the various cohorts in our study due to their enumeration at the sea entry traps, due to non-sampling of fish, poor quality DNA that could not be amplified, and poor scores at some loci for some fish, the number of fish in our pedigree is smaller than the number of fish in the productivity data (the productivity data being derived from the fish enumerated at the traps).

“L234- what is productivity in this sense? The ratio of recruits to spawners?”

Productivity is indeed the ratio of recruiting adults produced by a given number of spawners. We have clarified this point in the revised manuscript at lines 224-225 by saying *“providing us with a ratio of adult recruits per spawner, giving us an adult-to-adult productivity measure.”*

“L239- how is this calculated? Fixed number of eggs/kg Females?”

The fecundity of Irish Atlantic salmon can be calculated for both female grilse and MSW fish using a non-linear relationship as described in de Eyto et al. 2015 (doi:[10.1016/j.fishres.2014.11.017](https://doi.org/10.1016/j.fishres.2014.11.017)). Due to having better weight than length data for Burrishoole salmon, our fecundity estimates were calculated using a variation of this relationship. We have made this more explicit in the main text of the revised manuscript and refer the reader to the Supplementary Material that describes the process in more detail. See lines 227-231 of the revised manuscript where we now say, *“Since female MSW salmon and female grilse differ in their fecundity⁴⁹, variation in the annual grilse:MSW ratio could affect productivity if estimated using ova deposition. To explore this, we calculated an alternative productivity measure that involved ‘converting’ adults into eggs, using an unpublished relationship between female length and fecundity (Marine Institute, unpublished; Text S2).”*

“L242- fitness should be addressed in the introduction more explicitly”

We have brought fitness more to the fore in the Introduction, with explicit mention at lines 98-99 (*“Indeed, the lifetime fitness of released individuals relative to wild individuals is rarely directly measured.”*),

“L250- must be more clear about how this was accounted for, briefly summarise rather than must referring to another study

L252- unclear what is being permutated here.. in each year each individual is assigned a relative reproductive success based on the number of its children that return to spawn.. I don't see a Table S2 so I assume it is Table 2

Year | Individual ID | Provenance | RRS

1977 | 001 | Wild | 1.0

1977 | 002 | Hatchery | 0.3

1977 | 003 | hatchery | 0.2

Is this how the data look? A RRS for each individual for many years?”

The above two comments refer to how we calculated unbiased estimators of relative reproductive success, and how we tested for a statistical difference between these estimates. We would like to thank Referee 2 for highlighting the lack of clarity around our description of these analyses as this was something not immediately obvious to us as the authors of the manuscript. Referee 2's first point asks to include more detail about how to use the correction method of Araki & Blouin 2005 ([doi:10.1111/j.1365-294X.2005.02689.x](https://doi.org/10.1111/j.1365-294X.2005.02689.x)) to reduce bias in the estimates of relative reproductive success that arises from incomplete sampling and assignment error during parentage analysis. We have included a brief description of the correction method as follows: *“The method of⁵⁰ involved: (1) subtracting the number of offspring successfully assigned back to a parent from the number of offspring sampled; (2) dividing this difference by the total number of potential parents; (3) multiplying by the pedigree-derived assignment error, divided by one minus the assignment error and; (4) subtracting the result from the arithmetic mean LRS. This was done separately for each provenance, with the result being an unbiased estimate*

of mean LRS for both wild- and captive-bred salmon. Dividing captive-bred mean LRS by wild-bred mean LRS yielded the unbiased RRS estimator.” See lines 244-250.

Referee 2’s second point refers to what the permutation tests are actually testing, that is, what data are permuted and what entity is compared in each test. The data are not in the format as suggested by Referee 2. Each individual fish in our data has a unique ID, a genetic sex identification, provenance (captive-/wild-bred,), absolute lifetime reproductive success (W), and a Year of Spawning. An example is as follows:

ID	Sex	Provenance	W	Year of Spawning
BURR_FC_PSG_P_79_40	F	FC	2	1979

What is permuted is the difference in mean LRS between the two provenances. This process has been shown to be mathematically equivalent to testing if the relative reproductive success of captive-bred fish is different from one (one being, by definition, the fitness of the wild-bred fish). The mathematical proof of this theorem is given in Appendix II of Araki and Blouin 2005. Therefore, by finding a significant difference between mean LRS for each of the provenances, it follows axiomatically that the difference in relative reproductive success is significant. As suggested by Referee2, we have provided more detail on this process in the Methods section of the revised manuscript at lines 252-261 *“Separate one-tailed permutation tests were used to test for significant differences in mean LRS between captive- and wild-bred salmon for each of the six cohorts. The permutation tests generated 1,000,000 estimates of the difference in arithmetic mean LRS between captive- and wild-bred fish. P-values for each test were calculated as the proportion of the permuted samples greater than the observed difference in mean LRS between the provenances. One-tailed tests were chosen as we had an a priori expectation for lower LRS in captive-bred fish based on previous work. Permuting the difference between mean LRS estimates is mathematically equivalent to testing if the RRS of captive-bred fish is different from one - one being the relative fitness of wild-bred fish.”*

Finally, Referee 2 states that they could not find Table S2. Table S2 (now Table S3 in the revised Supplementary Materials) was submitted for peer review as part of the Supplementary Materials document.

“Table 2- unclear why these are not separated for wild and hatchery?”

Table 2 (now Table 1 in the revised manuscript) provides the overall comparison of relative reproductive success between wild and hatchery Atlantic salmon (column labelled “Overall”). Since RRS is a relative measure, giving them not separated is more convenient. Since LRS is an absolute measure of fitness and the focus of this paper is on the relative difference in fitness between wild- and captive-bred fish, we chose to report the mean LRS of each provenance in Table S3 of the revised Supplementary Materials. Also, our RRS measure involved correcting for potential biases arising from incomplete sampling and assignment error, as per the Araki and Blouin 2005 method, so RRS here would not exactly equal mean absolute LRS of captive divided by mean absolute LRS of wild. Hence, we would rather not include the annual mean absolute LRS values of each provenance in our new Table 1, to avoid confusing the reader.

Table 2- why are there some RRS values >1? Wild fish are assigned a value of 1, so I can't figure out how values could exceed this”

We have made Table 2 from the original manuscript into Table 1 in the revised manuscript. RRS is a ratio of the arithmetic mean LRS between two groups. We *a priori* predicted that captive-bred fish would have lower LRS and, hence, RRS values of <1. This prediction makes any statistical tests of the hypothesis inherently one-tailed. But in a single cohort (1977), captive-bred fish actually did better than wild-bred fish (RRS > 1), although the difference was not statistically significant. For consistency, the mean LRS of captive-bred fish was always the numerator in the RRS ratio, which is what led to these RRS values greater than 1 in that single year.

“L260- the reasoning behind the permutation tests and Fishers test are not well explained to me here”

We agree with Referee 2 that we did not clearly explain the reasoning behind the permutation tests. Fisher's Test was used to combine the *p*-values from each of the cohort-specific permutation tests in order to get an overall *p*-value to provide statistical

support for an *overall* difference in RRS across all cohorts. We have now explained the use of the tests more clearly in the revised manuscript at lines 261-265: *“To assess evidence for the hypothesis that, across all cohorts, there was an overall pattern of captive-bred salmon displaying lower LRS than wild-bred salmon, we combined the p-values from each of the six permutation tests using Fisher’s Combined Probability Test (FCPT). FCPT relies on taking the natural logarithms of the permuted p-values.”*

“L282- this is a clever idea to exclude non-spawners, I like it”

No response required, thanks. We are glad this method was deemed appropriate.

“L319- number of knots allowed in the smoother could make a large difference in the GAM, was this specified in any way?”

We did not specify the number of knots used in the smoothing term of the GAM in the original submission of this manuscript. We have done so in the revised manuscript at line 319. In order to satisfy ourselves that the number of knots chosen did not in fact change the results, we reran the GAM twice, specifying 19 and 39 knots in the smoothing term of each of these model runs. Variation in knot number made no qualitative difference in the results.

“L320- really interesting idea, why the residuals act as a density-corrected productivity metric is not entirely obvious”

The residuals can be viewed as a density-independent measure of recruits per spawner, that is to say, any non-linearity in the relationship between number of recruits and number of spawners due to density-dependence is removed. When the residuals are used as a productivity measure, variation in productivity due to some other explanatory variable is now independent of any underlying variation that would have arisen from an effect of population density on population productivity (e.g. due to non-linearity in the stock-recruitment relationship). While this is a common analytical technique in fisheries science, we now provide an explanation of this process in the revised Methods so as to

improve clarity of the manuscript for researchers not working in fisheries science. See lines 319-321: *“The residuals from this model are equivalent to a density-corrected productivity measure, as they represent a density-independent stock-recruitment relationship.”*

“Figure S1- not including the supplementary figures in this document makes the review quite challenging”

Figure S1 was provided in the Supplementary Materials document that was submitted for peer review. Apologies if that was not readily obvious.

“L409- is there potential bias in the ability to identify the origin of hatchery compared to wild parents?”

Misidentification is possible but was corrected for in the original pedigree construction project. Burrishoole captive-bred Atlantic salmon are genetically distinct from their wild-bred conspecifics and readily separate out in STRUCTURE analyses. The genotypic provenance (wild- versus captive-bred) of a fish was given priority over its phenotypic identification of being either wild- or captive-bred (based on being fin-clipped or not).

“L430- lots of density dependence literature for juvenile salmonids that could be referred to here- overall this is a great paragraph, very nicely explained!”

We have included reference to the work of Grossman and Simon (2020) and Jonsson et al (1998) on lines 422 and 425 respectively. We believe the addition of Grossman and Simon (2020) to be a particularly valuable inclusion in the references as it is a comprehensive review of density-dependence across salmonids, in general.

“L455- I read this as saying that wild fish were only sampled as kelts moving out after spawning, but we know there can be (and often is) lots of mortality after spawning, so I would expect that this would have a huge potential bias (with lots of variance among years)... wild fish are clearly less likely to be represented in the sample and offspring from

wild parents are less likely to have an assigned parentage (if I understand correctly).. it is quite crucial to explain in more detail why this is not a problem”

We address this issue with the inclusion of the following paragraph at lines 204-210 in the revised manuscript: *“The discrepancy in when each provenance was sampled (upstream versus downstream) could bias LRS estimates downward for wild-spawning, captive-bred fish, due to the higher cumulative in-river mortality risk inflating their expected number of zero LRS records relative to wild-bred parents. We explored this by comparing non-zero LRS records between the provenances. Mean LRS was still significantly lower for captive-bred relative to wild-bred fish, demonstrating that this source of bias was not driving the main results (in qualitative terms) from the analyses. See Text S1 for further details.*

Responses to Referee 3:

“The manuscript describes one of few studies have been able to estimate the lifetime contribution of captive-bred fish to subsequent generations relative to wild-bred fish. A key finding is evidence of transgenerational carry-over effects from the hatchery environment that affect the fitness of wild-bred offspring in Atlantic salmon. In my view, this represents an important study of broad interest to researchers whether their interests lie in salmonid fish ecology, the consequences of farm:wild animal interactions, population-restoration efforts, or conservation biology per se. It also represents an excellent example of a long-term study whose results have been helpfully and impressively informed by molecular genetic techniques.

We thank the reviewer for their very positive general appraisal of our MS and approach.

I have one 'major' point and a few minor comments for the authors' consideration.

MAJOR: Lines 213-214: The text reads "94% of wild-bred fish were tissue sampled as kelts in the downstream traps, while 95% of captive-bred fish were sampled pre-spawning in the upstream traps."

So, the wild-bred fish were captured after spawning had occurred while the captive-bred fish were captured before spawning had occurred. I may have missed this information, but how many days prior to and following spawning were the captive-bred and wild-bred fish, respectively, likely to have been captured?

Given life-history theory, ceteris paribus it is possible that the wild-bred fish were more likely to experience costs of reproduction and die prior to downstream migration as kelts. This would mean that the comparison of LRS between wild-bred and captive-bred salmon might include only the hardest of the post-spawning wild-bred salmon, whereas the upstream trap captured all captive-bred fish."

Given that this could favour any comparison of LRS towards wild-bred salmon (a key finding of the present ms), I feel that this potential caveat merits consideration and mention by the authors."

Referee 3 raises the same issue as Referee 2 does at Line 455, concerning the sampling of captive- and wild-bred salmon at different points in the lifecycle (pre- versus post-spawning). Our response to Referee 2, repeated below, is the same for Referee 3's question: we absolutely acknowledge this potential source of bias, and indeed tried to pre-emptively deal with it in the original manuscript, for example by devoting an entire section of additional supplementary text to it. In the revised manuscript, we have now clarified all this by inclusion of the following paragraph at lines 204-210: *"The discrepancy in when each provenance was sampled (upstream versus downstream) could bias LRS estimates downward for wild-spawning, captive-bred fish, due to the higher cumulative in-river mortality risk inflating their expected number of zero LRS records relative to wild-bred parents. We explored this by comparing non-zero LRS records between the provenances. Mean LRS was still significantly lower for captive-bred relative to wild-bred fish, demonstrating that this source of bias was not driving the main results (in qualitative terms) from the analyses. See Text S1 for further details."*

“Line 69 and elsewhere: The authors report differences in average LRS, but I wonder if they could not also report variation in LRS? Among other things, estimates of individual variability in LRS could inform estimates of effective population size, N_e .”

This is a good idea and we would like to thank Referee 3 for pointing it out. We have now calculated the variance in LRS, both on an overall and a cohort-specific basis for both provenances (lines 345-346), We provide point estimates of these variances in Table S4 of the Supplementary Material.

“Line 105: Replace “where” with “at which”

Manuscript edited as suggested at line 103.

“Line 116: Egg-to-smolt survival rates of 10% would be high, according to the data presented by Hutchings & Jones (1998. CJFAS). I think that their average estimate is less than 2%.”

This was an error. The published values for estimated egg to smolt survival for Atlantic salmon in the Burrishoole catchment ranges annually from 0.3% to 1.1% (Salmon Research Trust/Agency of Ireland including the Marine Institute 1970–2018). This has been corrected to reflect the above on line 114 of the revised manuscript.

“Lines 291 and 297: There is no such entity as ‘population fitness’. Fitness can only be defensibly applied to individuals. Please revise.”

A fair point, and indeed we agree that the term ‘population fitness’ is replete with conceptual problems. We have removed reference to ‘population fitness’ from the revised manuscript, replacing it with the term “*mean individual LRS*” on lines 292 and 298, to be more specific and avoid confusion with population vs individual fitness, absolute vs relative fitness, etc.

“Lines 316-317: I like this data-fitting approach.”

Thanks, glad this was deemed appropriate.